# The Application of Polycaprolactone Scaffolds with Poly(ε-caprolactone)–Poly(ethylene glycol)–Poly(ε-caprolactone) Loaded on Kidney Cell Culture

**DOI:** 10.3390/ma15041591

**Published:** 2022-02-20

**Authors:** Junyu Sun, Xinxin Liu, Zongrui Chen, Lin Jiang, Mingwei Yuan, Minglong Yuan

**Affiliations:** National and Local Joint Engineering Research Center for Green Preparation Technology of Biobased Materials, Yunnan Minzu University, Kunming 650504, China; sun__junyu@163.com (J.S.); 18845633851@163.com (X.L.); c1062767178@163.com (Z.C.); jianglin_1981@163.com (L.J.)

**Keywords:** cell culture, human embryonic kidney cells, PCEC coating, 3D PCL scaffold, 3D printing

## Abstract

Human embryonic kidney cells are the host of adenovirus type-5 (Ad5) amplification. An Ad5-vector-based COVID-19 vaccine has been proven to be tolerated and immunogenic in healthy adults. Therefore, a rationally designed scaffold for culturing human embryonic kidney cells is useful for further studying its mechanism of action. Herein, a three-dimensional layered reticulated polycaprolactone (PCL) scaffold coated with poly(ε-caprolactone)-poly(ethylene glycol)-poly(ε-caprolactone) (PCEC) was developed to proliferate human embryonic kidney cells and to be used to amplify the Ad5 vector. The results indicate that PCEC improves the hydrophilicity and the cell culture ability of PCL cell culture scaffolds, resulting in a three times higher cell proliferation ratio of human embryonic kidney cells compared with those grown on bare PCL cell culture scaffolds. Meanwhile, the cytotoxicity test results showed that the scaffold material is noncytotoxic. This work provides an effective and scalable method for the in-depth study of adenoviruses.

## 1. Introduction

Adenovirus type-5 (Ad5) is a vital member of the group C cells of human adenoviruses [1]. Ad5 recombinant adenovirus vectors can effectively deliver targeted antigens to mammalian hosts and induce extensive and strong humoral and cellular immune responses. Ad5 is widely used as a recombinant and nonreplication vector for vaccine development and plays a significant role in the recent formulae of COVID-19 vaccines. Human embryonic kidney cells are the host of human adenovirus vector Ad5 amplification [2,3]. The culture of human embryonic kidney cells is critical to the development and utilization of Ad5.

Cell culture technology is the key technology in biomedical research, tissue engineering, regenerative medicine, and biosensing techniques [4,5,6,7]. Cell cultures are helpful for revealing the basic biophysical and molecular mechanisms, which is the key technology for research on cell traits and the development of cell products [8,9,10]. Traditionally, most cell culture is realized on two-dimensional (2D) scaffolds [11,12]. Although this method is convenient, easy to use, and low in cost, the 2D cell culture technology affects the expression of genes and the signal transmission of cells in the organism, resulting in the gradual loss of the biological function of the cultured cells in the organism and the inaccurate expression of cells in vitro [13,14,15]. Therefore, the employment of three-dimensional (3D) structures and more realistic biochemical and biomechanical microenvironment cultures has attracted great research interest in order to provide cell growth environments that are similar to the biological environment, which is of great significance for cell cultures [16,17,18,19]. Suguru Nitta et al. have reported the spherical iHep, which delivers a greater genetic similarity to live liver tissue than cells cultured via 2D monolayer technology. This system was able to maintain the drug metabolism of canine hepatocytes in vitro, improving drug evaluation research [20]. Vaiyapuri Subbarayan Periasamy et al. successfully adhered and proliferated human mesenchymal stem cells (hMSCs) on 3D cell culture scaffolds derived from plant leaves [21]. Qipeng Hua et al. applied 3D printed-porous microgel for lung cancer cell culture, considering the significant difference between 2D cell environments and tumor cells in vivo. The results demonstrate that the cells in porous microgel were more similar to those extracted by the existing gold standard in the ROCK–actin pathway [22]. Notably, 3D cell culture scaffolds could simulate the extracellular matrix in living tissues for cell adhesion, migration and growth, effectively improving the activity and biological characteristics of regenerated cells [23,24,25]. However, the natural extracellular matrix (ECM)-derived scaffold suffers from high expense and low yield. Thus, the development of artificial scaffolds is necessary for the further investigation of adenoviruses [26]. Polycaprolactone (PCL) is a semi-crystalline aliphatic polyester material, which is an aliphatic polyester polymerized by a caprolactone monomer [27]. Compared with traditional synthetic scaffolds, PCL materials have high mechanical strength, convenient processing, and excellent biocompatibility, and their degradation products can be discharged from the human body [28,29,30,31,32]. However, PCL lacks cell affinity sites on the surface, which is not conducive to cell culture in vitro [33,34]. In this work, poly(ε-caprolactone)-poly(ethylene glycol)-poly(ε-caprolactone) (PCEC) was used to coat PCL using a fused deposition method (FDM) to regulate the hydrophilicity of PCL 3D-layered mesh scaffolds [35,36]. PCEC is a biodegradable triblock copolymer that can be easily prepared by a one-step ring-opening reaction of hydrophobic PCL and hydrophilic PEG [37]. The PCEC coating could effectively improve the hydrophilicity of PCL scaffolds, providing sufficient hydrophilic points to improve the cell affinity. Characterization of its physicochemical properties indicated that PCEC improves the hydrophilicity and the cell culture ability of PCL cell culture scaffolds, resulting in a three times higher cell proliferation ratio of human embryonic kidney cells in comparison with those grown on bare PCL cell culture scaffolds. Meanwhile, the cytotoxicity test results showed that the scaffold material is noncytotoxic. This work provides an effective and scalable method for the in-depth study of adenoviruses.

## 2. Materials and Methods

### 2.1. Preparation of Coating Materials

PCL–PEG–PCL (PCEC) copolymers were synthesized according to previous reports [38]. A certain amount of caprolactone and polyethylene glycol-6000 was taken and dried in a vacuum oven at 45 °C for 24 h. The dried caprolactone and polyethylene glycol with a mass ratio of 9:1 were mixed in a round-bottom flask. The catalyst stannous octanoate (Sn(Oct)_2_), with a mass fraction of 0.1%, was diluted with a certain amount of toluene before it was added into the round-bottom flask with a 1 mL syringe. Afterwards, the mixture in the round-bottom flask was devolatilized at 60 °C for 1 h to remove the toluene. Subsequently, the mixture in the round-bottom flask was heated in a blast oven at 140 °C for 8 h under a nitrogen atmosphere. After cooling to room temperature, the solid product was dissolved in dichloromethane and then slowly poured into tert-butyl methyl ether for dispersion and filtration. The coating product was obtained after drying at 45 °C. According to the GPC results, the macromolecular weight of M_n_ was 9490, of M_w_ was 11,336 and of PDI was 1.19.

### 2.2. Preparation of 3D Cell Culture Scaffold

The 3D cell culture scaffold was prepared from polycaprolactone (PCL) with molecular weights of M_n_ 54,700, M_w_ 98,000 and PDI 1.79. The PCL cell culture scaffold was prepared according to the 3D printing parameters in Table 1. PCL was melted into discs by FDM at 120 °C. The discs were constructed with four levels of stacked fibrous scaffolds with filaments arranged in parallel at the same layer and intersecting at adjacent layers. The discs had a diameter of 2 cm and a thickness of 0.1 cm, and the average filament diameter was 150–500 µm. The spacing between two adjacent layer was 250–300 µm. The PCEC coating was prepared by dissolving PCEC in a certain percentage of organic solvent consisting of methylene chloride and ethyl acetate, and the mixture was added to a high-precision spray gun and uniformly sprayed onto the surface of the scaffolds. The average coating content on the surface of the scaffolds was approximately 40 mg. After the sprayed scaffold was dried in a vacuum oven at 45 °C for 2 h, the in vitro cell culture scaffold with coating material could be obtained.

### 2.3. Characterization Methods

#### 2.3.1. X-ray Diffraction Analysis

The PCL cell culture scaffolds and the coated PCL scaffolds were taken. An X-ray diffraction analyzer (Bruker D8 ADVANCE A25X, BRUKER AXS GmbH, Karlsruhe, Germany) was employed to analyze the diffraction of the two samples at a scanning speed of 12° min^−1^, and the crystal structure was detected in the range of 10° to 90°.

#### 2.3.2. Morphology Analysis

The scaffold material of a certain size was prepared and sprayed with gold. The microstructure morphology of the scaffold section was observed and analyzed using an SEM (NOVA-NANOSEM-450, FEI, Hillsboro, OR, USA) instrument at a 5 kV acceleration voltage.

#### 2.3.3. Porosity Determination

The porosity of the scaffolds (*n* = 3 per sample) was assessed using the liquid replacement method. Since absolute alcohol easily penetrates the pores of 3D scaffolds without dissolving the coatings and scaffolds, absolute alcohol was used as a substitute for solution. The vacuumed scaffold was placed in a measuring cylinder filled with ethanol solution, and the volume was recorded as V_1_. The scaffold was placed in the measuring cylinder and sealed with a sealing film. The volume was recorded as V_2_ after standing for 20 min. Finally, the scaffold was removed from the measuring cylinder, and the remaining absolute alcohol volume in the cylinder was recorded as V_3_. According to Equation (1), the porosity of the scaffold can be obtained:(1)$$\mathrm{porosity}=\frac{V_{3}-{\;V}_{1}}{V_{2}-{\;V}_{3}}\;\times \;100\%$$

#### 2.3.4. Water Contact Angle of the Material

The water contact angle of the material was tested using the contact angle tester (JY-PHa, Chengde Jinhe Instrument Co., Ltd., Hebei, China). At room temperature, 4 μL deionized water was dropped on the surface of the material, and the equilibrium condition was photographed when the solid–liquid–gas three-phase interface was balanced. The contact angle was measured using the contact angle measurement method provided by the software ImageJ. In order to express the hydrophilicity more accurately, the water contact angle is usually recorded as the average value of the measured contact angle of three parallel samples.

The hydrophilic criterion is as follows. When the contact angle θ = 0°, the material surface is completely wet. When θ < 90°, the surface of the material is partially wet and the material has good hydrophilicity. A water contact angle of θ = 90°is the dividing line of wetting or not. When θ > 90°, the material surface is not wet and the material is hydrophobic material. When θ = 180°, it is completely unwetted.

#### 2.3.5. Determination of Water Absorption Rate

The prepared PCL cell culture scaffolds were cut into small blocks and dried in an air dryer at 45 °C for 24 h, and the weight of scaffolds was recorded as W_1_. The scaffold with a mass of W_1_ was immersed in PBS buffer solution at 37 °C for 24 h. Subsequently, the material was taken out, and the liquid on the surface was quickly wiped off and weighed, denoted as W_2_. Each process was repeated three times. The water absorption of the material was calculated according to Equation (2):(2)$$\mathrm{water}\;\mathrm{absorption}\;\mathrm{rate}=\frac{W_{2}-{\;W}_{1}}{W_{1}}\;\times \;100\%$$

#### 2.3.6. Determination of Degradation Rate

The scaffold material was dried in oven at 45 °C for 24 h, weighed and recorded as W_1_. The scaffold was placed in PBS buffer solution (37 °C, pH 7.4) and hydrolyzed in vitro for 12 weeks. The scaffold was taken out every three weeks after freeze-drying and recorded as W_2_. The in vitro degradation rate of the scaffold can be described by Equation (3):(3)$$\mathrm{degradation}\;\mathrm{rate}=\frac{W_{2}-{\;W}_{1}}{W_{1}}\;\times \;100\%$$

#### 2.3.7. Cell 3D Inoculation and Culture Process

Prior to cell inoculation experiments, two protoplast scaffolds with a diameter of 2 cm and a thickness of 1.2 mm were soaked in 70% ethanol for 12 h, sterilized by UV irradiation, washed twice with PBS buffer and finally placed in 6-well plates. Human embryonic kidney cells with a density of 7.57 × 105 mL^−1^ were inoculated into 6-well plates with scaffold samples of 0.17 mL per well. The 6-well plates were incubated in DMEM (4.0 mL L-glutamine, 100 U mL^−1^ penicillin, 100 µg mL^−1^ streptomycin) containing 10% FBS at 37 °C in a saturated humidity incubator with 5% CO_2_. Firstly, changes in cell numbers on PCL cell culture scaffolds and PCEC–PCL cell culture scaffolds were observed at 1, 3, 5 and 7 h after inoculation, and the cells cultured were photographed after 24 h under white light. The number of cells cultured on PCL cell culture scaffolds and PCEC–PCL cell culture scaffolds within four days was recorded, and the living cells in the whole process were counted.

#### 2.3.8. Live Cell Count

The cell suspension was diluted to 200–2000 cells mL^−1^ with 10% fetal bovine serum medium (the general dilution multiple was 100 times). Then, 0.1 mL of 0.4% trypan blue solution was gently mixed with 0.1 mL of cell suspension. After several minutes, the cells were counted with 10% fetal bovine blood cell counter.

#### 2.3.9. Cytotoxicity Test of Materials

The cytotoxicity of the material was qualitatively analyzed using the direct contact method and quantitatively analyzed with the MTT method.

##### Direct Contact Method

Direct contact method: A 2 mL cell suspension (1 × 105 cells mL^−1^) was inoculated in a 35 mm culture dish. The culture dish was placed in a cell incubator for 24 h until monolayer confluence and then was replaced with 0.8 mL of fresh medium.

PCL cell culture scaffolds and PCEC–PCL cell culture scaffolds were placed in each culture dish as test samples, and a blank control, positive control (polyurethane (ZDEC) diethyl dithiocarbamate) and negative control (high-density polyethylene (PS)) were set up. At least two parallel samples were set up for each sample. All cultures were placed in a cell incubator at 37 ± 1 °C containing 5 ± 1% CO for at least 24 h. The cultured cells were stained with trypan blue reagent, and the cell morphology of each test sample, blank control, negative control, and positive control were observed using a microscope.

##### MTT (3-(4,5)-Dimethylthiahiazo (-z-y1)-3,5-Di-Phenytetrazoliumromide) Cytotoxicity Test

Three scaffolds, including PCL cell culture scaffolds, PCEC–PCL cell culture scaffolds and a blank experiment, were placed in 6-well plates. PCL cell culture scaffolds were labeled as experimental group 1, PCEC–PCL cell culture scaffolds were labeled as experimental group 2 and the blank experiment was noted as the control group. Cells in the log phase were collected and counted. Then, the concentration of cell suspension was adjusted by inoculating 2 mL of cell suspension in each well. After the number of cells reached 20 × 10^4^ cells per well, all cultures were placed in a cell incubator containing 5 ± 1% CO_2_ at 37 ± 1 °C for at least 24 h. At the end of the incubation period, the original culture medium was replaced with 2 mL of culture medium containing 0.5 mg·mL^−1^ of MTT, continuing the incubation period for another 4 h. Afterwards, the culture medium containing MTT was removed and 1 mL of dimethyl sulfoxide was added, and the mixture was shaken for 10 min. The above cells were transferred into a 96-well plate and its absorbance value was measured at 570 nm using an enzyme marker. The blank experiment was used as the control group, and the survival rate of cells in the positive control group was labeled as 100%. Herein, the survival rate of the other parallel experimental groups can be calculated as OD (blank group)/OD (experimental group) × 100%.

## 3. Results

### 3.1. Physical and Chemical Properties of Scaffold

#### 3.1.1. XRD Characterization of Cell Culture Scaffolds

Figure 1 shows the X-ray diffraction pattern of PCL cell culture scaffold material recorded at a scanning speed of 12° min^−1^, where (a) shows the diffraction pattern of PCL cell culture scaffolds and (b) shows that of PCEC–PCL cell culture scaffolds. It can be seen in Figure 1a that the main diffraction peaks are located around 22° and 24°, indicating the crystalline plane of the PCL material. In Figure 1b, the position of the diffraction peak of the PCEC–PCL cell culture scaffolds is consistent with that of the PCL material, indicating that the coating material did not destroy the original crystal structure of PCL, which was conducive to maintaining the original properties of the PCL material. The peak intensity in Figure 1a is notably stronger than that in Figure 1b. This phenomenon should be attributed to the successful loading of the coating on the surface of PCL cell culture scaffolds, which reduces the X-ray diffraction intensity of PCL materials.

#### 3.1.2. SEM of Scaffold Material

Figure 2 shows the synthetic strategy of PCL cell culture scaffolds and the surface morphology before and after loading the coating. Figure 2a shows a sketch of the preparation of a PCL scaffold. The PCL material was melted in the heating nozzle of the FDM printer before being extruded from the nozzle. According to the requirement mentioned above, the melted PCL was constructed into a specific shape under the control of the computer program (Figure 2b–d) are SEM images of PCL cell culture scaffolds and PCEC–PCL cell culture scaffolds, respectively. It can be seen that the 3D printed PCL cell culture scaffolds have a crisscross layered structure. After PCEC is used to coat the surface of the PCL, the scaffolds maintain their cross-layered morphology with a rough coating. Obviously, the coating material on the surface of the PCL cell culture scaffolds does not affect the structure of the scaffolds. The pore connectivity rates of the PCL cell culture scaffolds and the PCEC–PCL cell culture scaffolds were 100%.

#### 3.1.3. The Porosity of the Scaffolds

The porosity data of the two cell culture scaffolds obtained using the liquid replacement method are shown in Figure 3. The porosity of the PCEC–PCL cell culture scaffolds, shown in columns a, b, and c, is 30.80% ± 0.27%. The porosity of the PCL cell culture scaffolds, shown in columns a_1_, b_1_, and c_1_, is 37.61% ± 0.36%. Obviously, the porosity of the PCL cell culture scaffold decreases after coating, which is attributed to the formation of a coating that increases the diameter of the scaffold fiber, decreasing the porosity of the PCL cell culture scaffold. However, according to the SEM images, there is still sufficient space between two adjacent fiber scaffolds in the PCEC–PCL cell culture scaffolds, facilitating the attachment and growth of cells.

#### 3.1.4. Hydrophilicity of the Scaffolds

The water contact angles of the PCL cell culture scaffolds and PCEC–PCL cell culture scaffolds were about 137.6° and 62.7°, respectively. This indicates that the hydrophilicity of the PCL cell culture scaffolds improved significantly after loading PCEC. The modification of hydrophilicity is conducive to the attachment and growth of cells on the scaffolds.

#### 3.1.5. Water Absorption Rate of the Scaffolds

The water absorption data are shown in Figure 4. The PCL cell culture scaffolds and PCEC–PCL cell culture scaffolds were immersed in buffer solution for 24 h. The average water absorption rates of the PCL cell culture scaffolds were calculated to be 6.19% ± 0.41% according to columns a, b and c. The average water absorption rates of the PCEC–PCL cell culture scaffolds were 9.77% ± 0.23%, according to columns a_1_, b_1_, and c_1_, and their water absorption rates are notable. The water absorption of PCL cell culture scaffolds was significantly improved after loading the PCEC due to the strong water absorption of the coating material. The uniform loading of the PCEC coating on the surface of the PCL cell culture scaffolds improves the water absorption and increases the active sites on the PCL surface, facilitating the attachment and reproduction of cells.

#### 3.1.6. Determination of Scaffold Degradation Rate

PCL cell culture scaffolds and PCEC–PCL cell culture scaffolds were immersed in buffer solution with a constant temperature of 37 °C for 12 weeks, and the relationship between the degradation rate and degradation time was recorded. As shown in Figure 5, the degradation rates of the PCL cell culture scaffolds were 0.81% ± 0.05%, 1.19% ± 0.03%, and 1.57% ± 0.1% in the 4th, 8th, and 12th weeks, respectively. The degradation rates of the PCEC–PCL cell culture scaffolds were 1.06% ± 0.03%, 1.39% ± 0.13% and 1.82% ± 0.05% in the 4th, 8th, and 12th weeks, respectively. It can be seen that the degradation rate of PCL cell culture scaffolds without a coating was relatively slower than the degradation rate of the PCEC–PCL cell culture scaffolds in vitro. According to the results above, the PCEC coating changed the interface properties of the PCL cell culture scaffolds, improving the degradation effect of PCL.

### 3.2. Cell Culture Results

#### 3.2.1. Initial Stage of Cell Inoculation

Figure 6a–d are microscopic images of cell adhesion at different magnified scales on PCEC–PCL cell culture scaffolds and PCL cell culture scaffolds, respectively, after inoculation for 24 h. Clearly, there are cells adhering to the surface of both scaffolds at the same incubation time, but the blank area on the PCL cell culture scaffolds is larger than that on the PCEC–PCL cell culture scaffold. This phenomenon indicates that PCL and PECE coatings can be used as scaffolds for the growth of cells, but the growth rate is correlated with the properties of the materials. As mentioned above, the hydrophilicity of the PCEC-coated PCL scaffold is better than that of the pure PCL scaffold, providing a better environment for cell attachment and growth. This also provides evidence of the good cytocompatibility of the PCEC material.

The proliferation of human embryonic kidney cells on PCL cell culture scaffolds and PCEC–PCL cell culture scaffolds at the early stage of inoculation is shown in Figure 7. After inoculation on the PCL cell culture scaffolds or the PCEC–PCL cell culture scaffolds, the number of cells adhering to the coated cell culture scaffolds increased significantly in comparison with that of the PCL cell culture scaffolds. In addition, the ratio of cell number to time on the PCEC–PCL cell culture scaffolds is larger than that on the PCL cell culture scaffolds in Figure 7, further confirming the results that PCEC-coded PCL scaffolds are beneficial for the growth and proliferation of human embryonic kidney cells.

Table 2 describes the number of cells harvested and the corresponding proliferation multiples of the two cell culture scaffolds after culturing for four days. According to the proliferation data in the table, the number of cells harvested after four days from PCL cell culture scaffold was less than that harvested from the PCEC–PCL cell culture scaffolds. The average proliferation rate of the PCL cell culture scaffold was about 1.924 ± 0.125 times, while the average proliferation rate of the PCEC–PCL cell culture scaffolds was 5.152 ± 0.125 times, indicating that the coating of PCEC on PCL scaffolds significantly improved the cell culture ability of the scaffolds.

Figure 8a–c show the cell inoculation results of the PCL cell culture scaffolds, where c is the bottom of the cell culture dishes observed after culturing. Obviously, the cells were exfoliated from the scaffolds due to the poor ability of cells to adhere to the scaffold surface. Figure 8d–f show the cell inoculation results on the PCEC–PCL cell culture scaffolds, where (f) is the bottom of the cell culture dishes. It can be observed that few deciduous cells disperse in the selected area, implying the good adhesion of PCEC to cells. Therefore, coating PCEC on PCL greatly improves the cell culture ability of the scaffold. Moreover, the hydrophilic properties of the PCL cell culture scaffolds were also improved due to the coating material providing a stable environment for cell growth and proliferation. The successful cultivation of human embryonic kidney cells on PECE-modified PCL cell culture scaffolds is helpful for studying human embryonic kidney cells resembling the cell survival environment.

#### 3.2.2. Cytotoxicity Test Results

##### Direct Contact Test Results

In order to verify the cytotoxicity of the cell culture scaffolds, including the blank control groups, positive control groups and negative control groups, an analysis was initially conducted as a reference. As shown in Figure 9a,b, the microscopic observations of the blank control groups of cells at different magnifications show normal cell morphology. Compared with the blank control group (b), the negative control group (f) exhibits normal cell morphology and an obvious toxic region, indicating that the negative control group is noncytoxic. On the contrary, the cells in the positive control group (d) show round shrinkage, loose adherence and an incomplete cell membrane, indicating that the material was cytotoxic. Based on the above phenomenon, the cytotoxicity of the PCL cell culture scaffolds and PECE–PCL cell culture scaffolds was determined. As shown in Figure 10a,b, both PCL cell culture scaffolds and PCEC–PCL cell culture scaffolds exhibited normal cell morphology with obvious toxic areas, similar to what appeared in the negative control group in Figure 9f, indicating that PCL cell culture scaffolds and the coating materials loaded on the PCL cell culture scaffolds are all nontoxic materials for cells and can be used as raw materials for the preparation of cell culture scaffolds.

##### MTT Test Results

A further cytotoxic experiment using the MTT method was conducted, and the results are shown in Table 3 and Figure 11. As mentioned in GB/T 16886.5-2017, a survival rate lower than 70% for the blank indicates the potential cytotoxicity of the test sample. The relative cell survival rate of the PCL cell culture scaffold extract was calculated to be 93.2%, while the relative cell survival rate of PCEC–PCL cell culture scaffold extract was 90.5%. Both samples exhibit a high survival rate in comparison with the blank samples, confirming the favorable cytotoxicity of PCL and PCEC–PCL cell culture scaffolds.

## 4. Conclusions

In this study, biodegradable PCL cell culture scaffolds modified with PCEC were successfully used as 3D layered cell culture scaffolds for culturing human embryonic kidney cells. PCEC, when loaded on to the PCL cell culture scaffolds, improved their hydrophilicity, providing more hydrophilic water points for the attachment and growth of cells. In addition, the proliferation multiple of human embryonic kidney cells on PCEC-coated PCL cell culture increased from about 2.47 × 10^5^ to about 6.62 × 10^5^ folds. A cytotoxicity test of the PCL and PCEC–PCL scaffold materials showed that they were nontoxic. Therefore, this work provides a new concept for the cultivation of human embryonic kidney cells, which is conducive to the execution of further research on adenoviruses.

## Figures and Tables

**Figure 1 materials-15-01591-f001:**
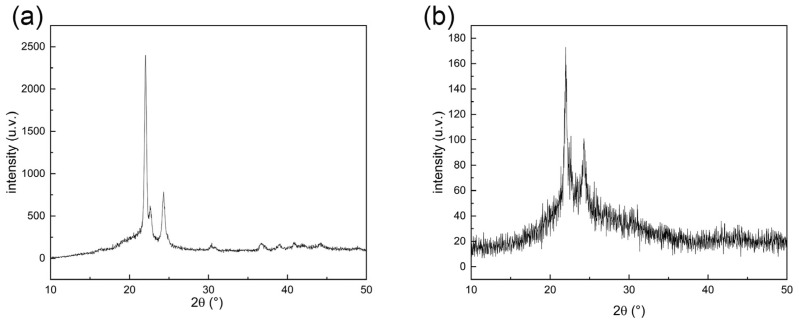
XRD patterns of scaffold materials: (**a**) PCL cell culture scaffolds, (**b**) PCEC–PCL cell culture scaffolds.

**Figure 2 materials-15-01591-f002:**
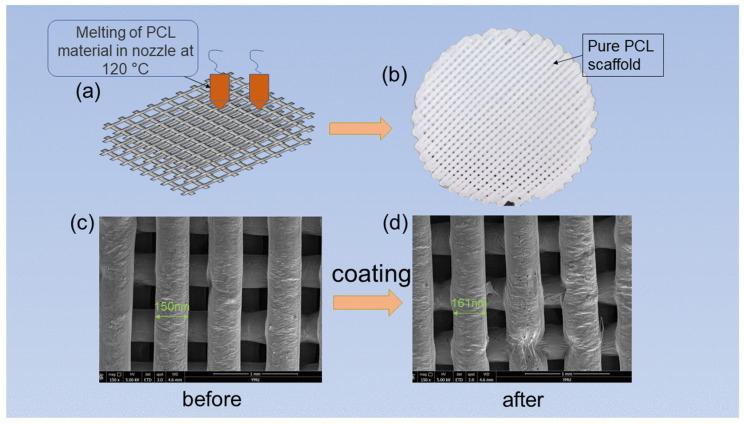
(**a**) Synthetic strategy of PCL cell culture scaffolds; (**b**) photograph of PCL cell culture scaffolds; (**c**) SEM image of PCL cell scaffold; and (**d**) SEM image of PCEC–PCL cell culture scaffold.

**Figure 3 materials-15-01591-f003:**
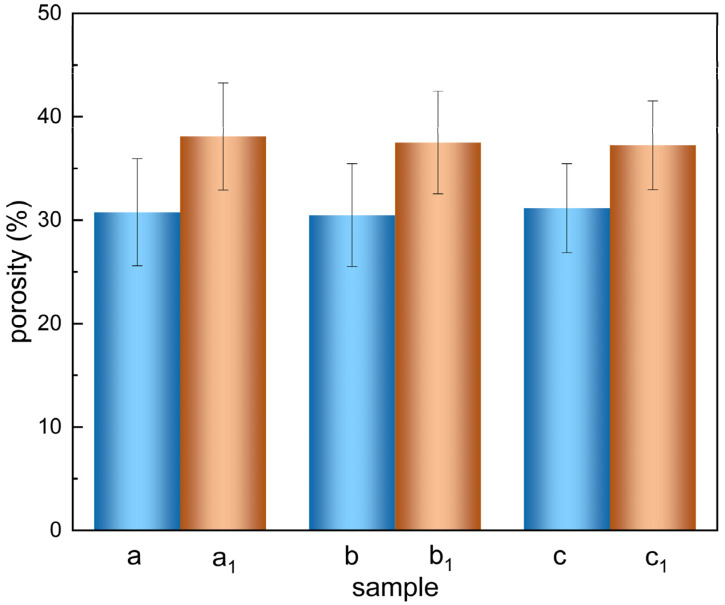
Porosity diagram of the PCL cell culture scaffolds and PCEC–PCL cell culture scaffolds.

**Figure 4 materials-15-01591-f004:**
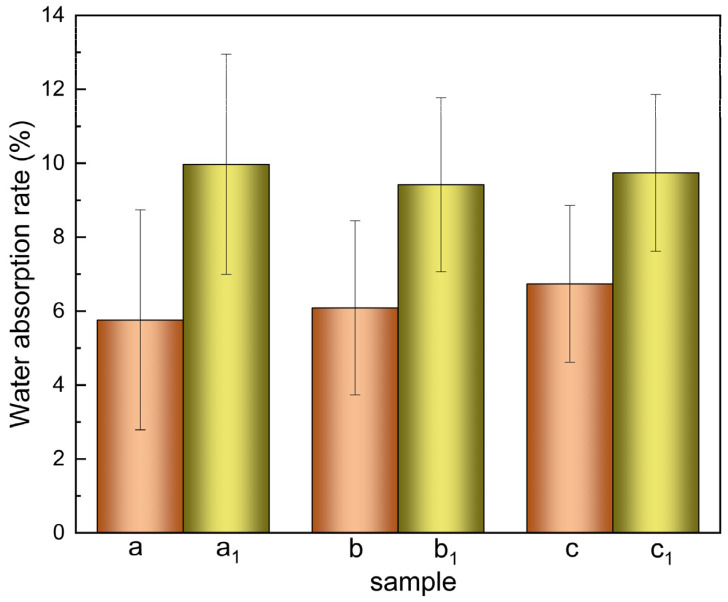
Water absorption rate diagram of PCL scaffold and PCEC–PCL cell culture scaffolds.

**Figure 5 materials-15-01591-f005:**
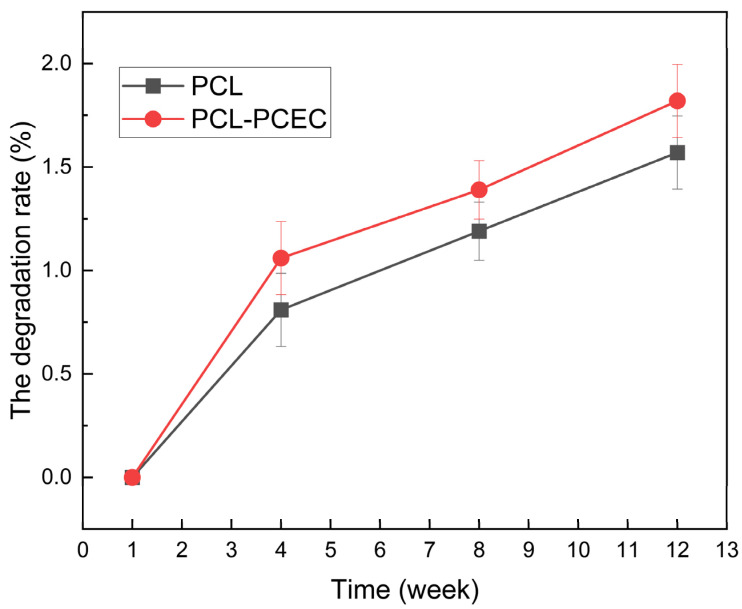
Degradation rate of PCL cell culture scaffolds and PCEC–PCL cell culture scaffolds (weeks 2, 4, 8, and 12).

**Figure 6 materials-15-01591-f006:**
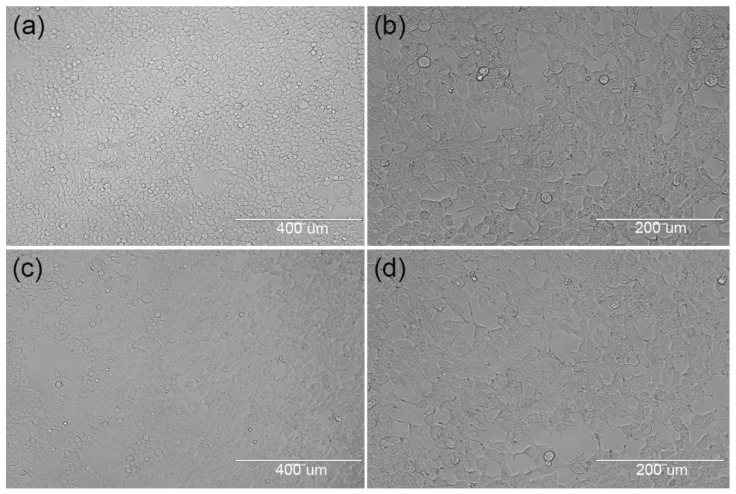
Microscopic images of cells adhering to each group of scaffolds after 24 h of culture; (**a**,**b**): distribution of cell adhesion on the surface of PCEC–PCL cell culture scaffolds; (**c**,**d**): distribution of cell adhesion on the surface of PCL cell culture scaffolds.

**Figure 7 materials-15-01591-f007:**
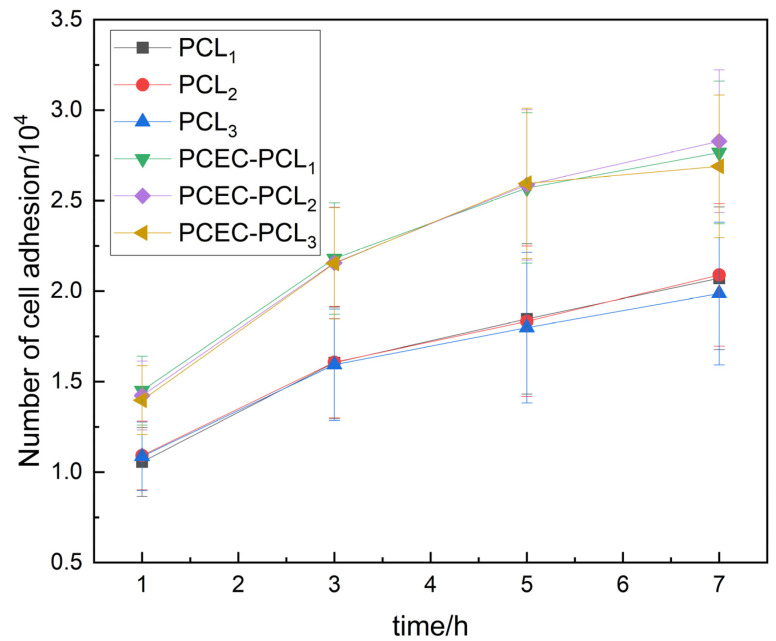
Trends in cell numbers within 7 h of cell inoculation.

**Figure 8 materials-15-01591-f008:**
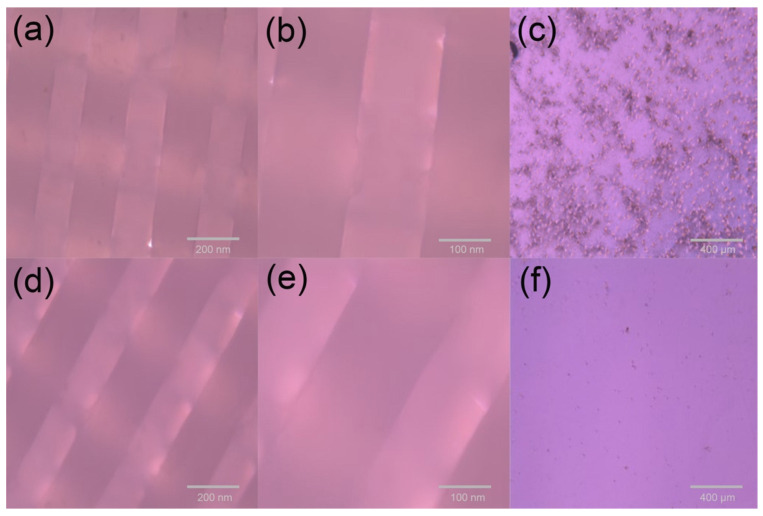
Cell inoculation results: (**a**,**b**) are microscopic images of cell culture results for PCL scaffolds at 200 nm and 100 nm magnification, respectivel. (**c**) The bottom of the PCL scaffold culture dish. (**d**,**e**) are microscopic images of cell culture results of 200 nm and 100 nm magnification of the PCEC–PCL cell culture scaffolds, respectively. (**f**) The bottom of the PCL scaffold culture dish.

**Figure 9 materials-15-01591-f009:**
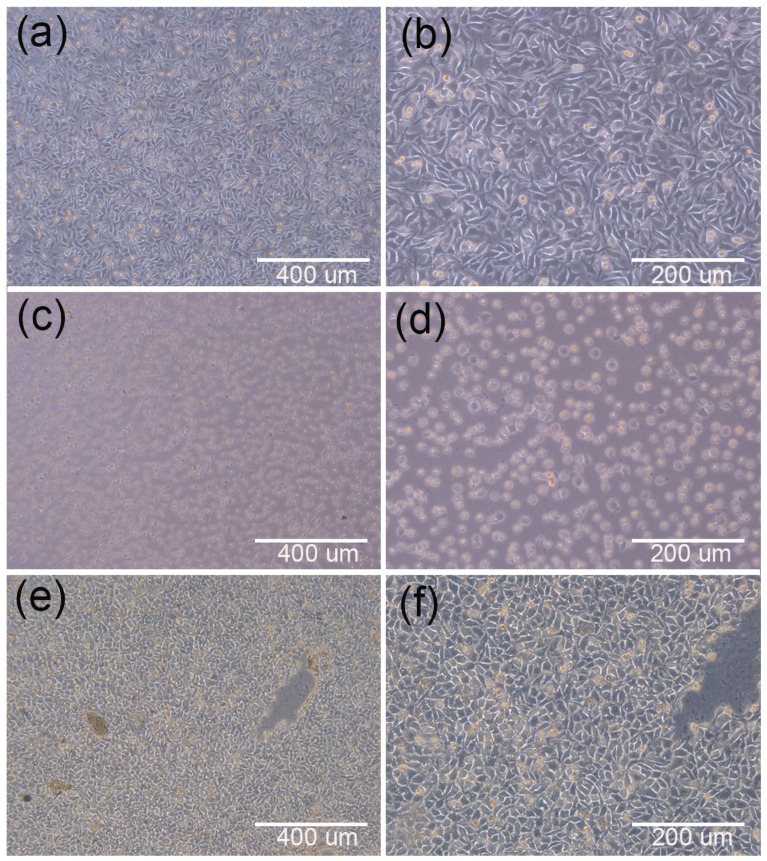
(**a**,**b**) show the blank control groups of cells; (**c**,**d**) show the cytotoxic positive control group; and (**e**,**f**) show the cytotoxic negative control group.

**Figure 10 materials-15-01591-f010:**
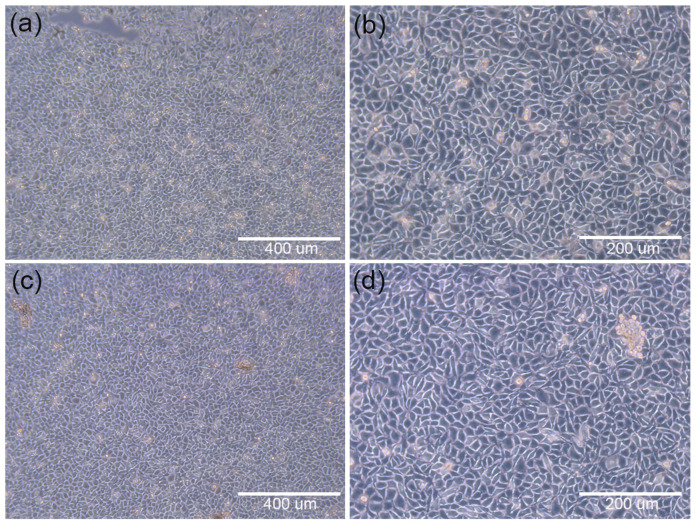
Microscopic images of each experimental group with 400 μm and 200 μm magnification: (**a**,**b**) PCL cell culture scaffold; (**c**,**d**) PCEC–PCL cell culture scaffolds.

**Figure 11 materials-15-01591-f011:**
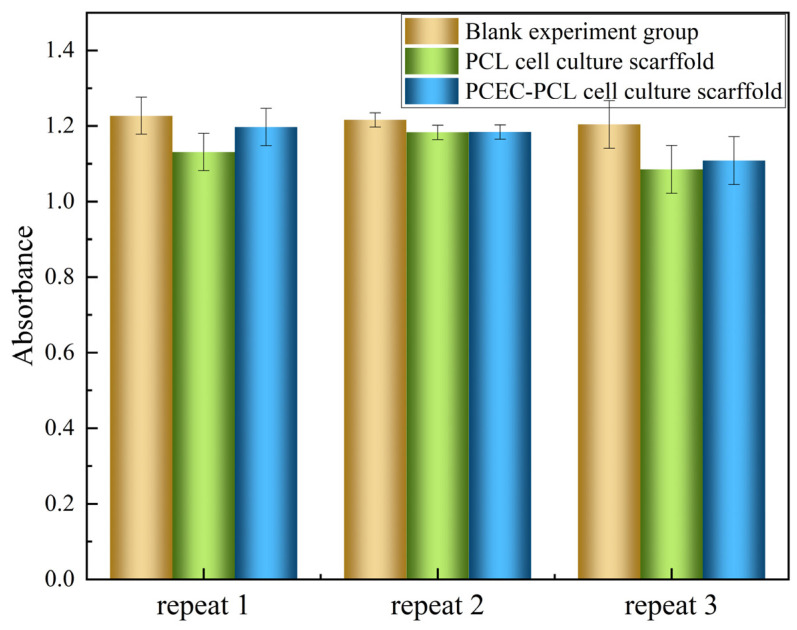
The OD (570) data obtained with the MTT method.

**Table 1 materials-15-01591-t001:** PCL 3D cell culture scaffold printing parameters.

Printing Parameter	Filling Rate (%)	Temperature (°C)	Speed (mm s^−1^)	Nozzle Diameter (mm^−1^)	Filling Spacing (μm)
/	100	120	35.0	1.0	250–300

**Table 2 materials-15-01591-t002:** Results of cell proliferation after four days of cell culture.

Sample	Parallel Samples	Harvesting Cell Count	Proliferative Multiple	Results
PCL	1	2.689 × 10^5^	2.1	Poor proliferation effect
	2	2.433 × 10^5^	1.9	
	3	2.305 × 10^5^	1.8	
PCEC-PCL	1	6.855 × 10^5^	5.3	Good proliferation effect
	2	6.6 × 10^5^	5.1	
	3	6.435 × 10^5^	5.0	

**Table 3 materials-15-01591-t003:** The OD (570) data obtained with the MTT method.

Sample Number	Absorbance (OD = 570)			Relative Cell Survival (100%)
	Repeat 1	Repeat 2	Repeat 3	
Blank experimental group	1.227	1.216	1.204	100
PCL cell culture scaffold	1.131	1.183	1.085	93.1743
PCEC–PCL cell culture scaffold	1.197	1.184	1.108	90.5428

## Data Availability

Data sharing is not applicable for this article.

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
