# Peer review of "The Application of Polycaprolactone Scaffolds with Poly(ε-caprolactone)–Poly(ethylene glycol)–Poly(ε-caprolactone) Loaded on Kidney Cell Culture"

_materials, 2022, doi:10.3390/ma15041591_

Round 1

Reviewer 1 Report

On request of Materials, I have revised the manuscript titled “The application of polycaprolactone scaffolds loaded poly(É›-caprolactone)–poly(ethylene glycol)–poly(É›-caprolactone) on kidney cell culture”, by Junyu Sun and co-authors.

Before expressing myself on the work, I would like to point out that, lacking the line numbers, it will be difficult for me to point out the different problems of this work in a precise way, and I will only be able to talk about them in general.

Notably, I ask the authors to pay more attention to the requests of the journals on which they would like to publish their works, to edit a manuscript that is suitable for them. Reading this work in fact, a reviewer is forced to say that it is not suitable to be published on Materials because it does not respect the journal’s requirements.

Anyway, the present work is based on the understanding that the adenovirus type-5 (Ad5) is amplified in human embryonic kidney cells, and that Ad5 vector-based COVID-19 vaccine proved to be tolerated and immunogenic in healthy adults. So, with the end of studying its mechanism of action, the main scope of this study was developing scaffolds for the proliferation of human embryonic kidney cells to be used to amplify Ad5 vector.

In this regard, by using biocompatible and biodegradable polycaprolactone (PCL) as polymer matrix, three-dimensional PCL scaffolds for kidney cells culture were prepared by melt deposition method, which were subsequently coated with PCEC terpolymer material to improve the physicochemical properties of PCL alone. According to the results, PCEC coating improved both the hydrophilicity and the cell culture ability of pure PCL scaffolds, while cytotoxicity was not observed.

Considering the worldwide worrying pandemic situation, which afflict humanity with periodic waves of infections and deaths since 2020, to study adenovirus and their variants in an ever more in-depth manner, to be able to develop increasingly effective and better tolerated vaccines, is needed.

In this regard, the present study, which addressed the preparation of scaffolds suitable for the proliferation of cells capable of amplifying Ad5, already recognized to be immunogenic and well tolerated by humans, may be of interest.

Unfortunately, despite the scientific relevance of the contents of this study, the quality of its form is very poor and significantly lowers its stature. The work is overall poorly written. English is not good. I recommend that authors have their work reviewed by a native English expert to reduce the various grammatical and typing errors. In this regard, authors are requested to present a certification attesting to this language revision work.

The template of Materials and its instructions have often not been respected. The work is not uniform either as regards the font or as regards the size of the characters.

Please, check all manuscript and, where necessary specify the abbreviations at their first mention.

In the abstract, replace tolerant with tolerated.

In the abstract, modify the repetitive expression “for researchers to research”.

Keywords. All without capital letters.

Please, check all manuscript and adapt the way of citing the references numbers to Materials template.

Please, check all manuscript and adapt the sections titles and the way of writing both the titles of sections and sub-sections to Materials template.

Please, check all manuscript and make the tenses of verbs uniform. I recommend using past tense but, in any case, not to use a mixture of past and present.

In 2.23. D Cell culture scaffold preparation, 2.23. is incorrect.

Please, reformat all Tables and mathematical formulas, which are frequently also not uniform in font and size, according to Materials template. The font of Tables and many sub-sections titles is incorrect.

Please, check all manuscript and remove the many nonsense spaces, as well as the many nonsense dots in the middle of sentences.

Please, check all manuscript and insert the Figure panels and related letters as indicated in the template.

Please, check all manuscript and remake all Figures using the Arial font for the writings.

All experiments must be performed in triplicate and results must be reported as means ± standard deviation. Please, correct the data in Tables, accordingly. Consequently, in all the bars graphs, the bars of error must be included. Please, correct. The same must be applied at dispersion graphs of Figure 5 and Figure 6.

On page 11, Figure 7 is erroneously dispersed throughout the text.

After conclusions, the authors must insert the parts requested by the Template using the correct font and characters size.

The references list does not respect the instructions and must be corrected.

I conclude by saying that the work is not very well cared for, and I advise against the publication in this form for Materials.

Reviewer 2 Report

This study aims to improve kidney cell attachment on PCL FDM scaffolds by coating the PCL fibre with PCEC block copolymer. The cytotoxicity of the PCL scaffolds with and without the PCEC coating was tested.

My comments to the authors are as follows:

  1. Please be more specific of the ratio of PCL and PEG in the block copolymer preparation (section 2.1).
  2. In section 2.23 (D Cell culture scaffold preparation), the FDM printing parameters should be improved for a more consistent fibre diameter (currently varies between 150-500 µm), maybe consider using a smaller nozzle diameter. The fibre spacing should be consistent. The improved printing parameters would produce 3D scaffolds with consistent dimension in different treatment groups, and the difference of cell attachment in the scaffold groups would be solely caused by the fibre surface properties.
  3. Also in section 2.23, the authors need to provide the molecular weight of PCL used and quantify the amount of PCEC coated on the PCL scaffolds.
  4. In section 2.2.7, please state the dimension of scaffold for cell seeding and culture.
  5. There are popular qualitative and quantitative in vitro cytotoxicity assays such as LIVE/DEAD and AlamarBlue assays that will enable statistical analysis of the cell viability and cell numbers on these 3D scaffolds.
  6. Figure 3 is missing figure legend.
  7. Figure 5 is missing statistics. Was the higher degradation in the coated scaffold group purely due to the fast degradation of the PCEC coating?
  8. Figure 6b and 6c, I’m not sure what are shown here. The cell morphology does not represent typical attached cells.
  9. A general comment, all microscope images are missing scale bars.
  10. To assess the morphology of kidney cells cultured on the scaffolds and compare it with that of the ones extracted by the gold standard, it is highly recommended to use histological techniques.

In conclusion, the authors have made an interesting attempt to improve the cell attachment on PCL scaffolds using a hydrophilic coating. However, to back up the statement of enhanced cell attachment on the PCEC coated scaffolds while maintaining their phenotype, the authors will need to improve the methods and study design.

Round 2

Reviewer 1 Report

Dear authors,

I have reconsidered your revised manuscript. Unfortunately, the work of revision you have done is not enough to make this paper ready for publication.

There are still many typos and grammar errors. As an example, only in the abstract:

line 13. "scaffolds", plural is incorrect.

line 14. "proliferable"????

line 15. "hydrohilicity" is incorrect.

Often in the manuscript you have not used the capital letters in the correct way. You should recheck all the manuscript and correct accordingly.

Examples:

In the abstract. "Poly-caprolactone" "Poly(ε-caprolactone)-poly(ethylene glycol)-poly(ε-capro-lactone)", not with capital letters.

The same in lines 64 and 83.

On the contrary the titles in the first rows of all the Tables should have capital letters. All the words in the headings and sub-headings should have the first letter as capital. On the contrary, the word in equations, without capital letters.

Practically, all the Figures' captions lack the dots at the end. The same in some Tables titles.

Between the numbering and the first word of all the sub-heaning dots are missing.

Line 280. Not in bold, please.

Line 286. There are useless spaces between brackets and numbers.

Importantly, I have asked authors to provide a certificate assessing the English revision by a native expert, which has not been given.

Reviewer 2 Report

Dear authors,

Thanks for the extra work to address my previous comments. I can see great improvement of quality in the manuscript.

Please find my minor comments in the pdf file attached.

Kind regards.
